# *Trypanosoma cruzi* DNA Polymerase β Is Phosphorylated In Vivo and In Vitro by Protein Kinase C (PKC) and Casein Kinase 2 (CK2)

**DOI:** 10.3390/cells11223693

**Published:** 2022-11-21

**Authors:** Edio Maldonado, Diego A. Rojas, Fabiola Urbina, Lucía Valenzuela-Pérez, Christian Castillo, Aldo Solari

**Affiliations:** 1Programa de Biología Celular y Molecular, ICBM, Facultad de Medicina, Universidad de Chile, Santiago 8380492, Chile; 2Instituto de Ciencias Biomédicas (ICB), Facultad de Ciencias de la Salud, Universidad Autónoma de Chile, Santiago 8910132, Chile; 3Facultad de Medicina Veterinaria y Agronomía, Universidad de Las Américas, Santiago 7500975, Chile

**Keywords:** *Trypanosoma cruzi*, DNA polymerase β, protein kinases, PKC, CK2, Wee1

## Abstract

DNA polymerase β plays a fundamental role in the life cycle of *Trypanosoma cruzi* since it participates in the kinetoplast DNA repair and replication. This enzyme can be found in two forms in cell extracts of *T. cruzi* epimastigotes form. The H form is a phosphorylated form of DNA polymerase β, while the L form is not phosphorylated. The protein kinases which are able to in vivo phosphorylate DNA polymerase β have not been identified yet. In this work, we purified the H form of this DNA polymerase and identified the phosphorylation sites. DNA polymerase β is in vivo phosphorylated at several amino acid residues including Tyr35, Thr123, Thr137 and Ser286. Thr123 is phosphorylated by casein kinase 2 and Thr137 and Ser286 are phosphorylated by protein kinase C-like enzymes. Protein kinase C encoding genes were identified in *T. cruzi,* and those genes were cloned, expressed in bacteria and the recombinant protein was purified. It was found that *T. cruzi* possesses three different protein kinase C-like enzymes named TcPKC1, TcPKC2, and TcPKC3. Both TcPKC1 and TcPKC2 were able to in vitro phosphorylate recombinant DNA polymerase β, and in addition, TcPKC1 gets auto phosphorylated. Those proteins contain several regulatory domains at the N-terminus, which are predicted to bind phosphoinositols, and TcPKC1 contains a lipocalin domain at the C-terminus that might be able to bind free fatty acids. Tyr35 is phosphorylated by an unidentified protein kinase and considering that the *T. cruzi* genome does not contain Tyr kinase encoding genes, it is probable that Tyr35 could be phosphorylated by a dual protein kinase. Wee1 is a eukaryotic dual protein kinase involved in cell cycle regulation. We identified a Wee1 homolog in *T. cruzi* and the recombinant kinase was assayed using DNA polymerase β as a substrate. *T. cruzi* Wee1 was able to in vitro phosphorylate recombinant DNA polymerase β, although we were not able to demonstrate specific phosphorylation on Tyr35. Those results indicate that there exists a cell signaling pathway involving PKC-like kinases in *T. cruzi*.

## 1. Introduction

American trypanosomiasis or Chagas disease is caused by the unicellular flagellated protozoan *Trypanosoma cruzi*, which affects over 8 million people worldwide [1,2]. This disease causes approximately 50,000 deaths each year and 70–100 million people are at risk of infection in endemic areas of Latin America [1,2,3].

Chagas disease has spread all over the world due to migration, which has brought the disease outside of the endemic areas and the transmission occurs vertically (congenital) and via blood and tissue/organ donations [2,4]. In endemic areas, the disease is transmitted by blood-sucking insects of the subfamily *Triatominae,* which can infect a mammalian host [5]. The *T. cruzi* parasite is an intracellular protozoan that can undergo a complex life cycle between the insect vector and a mammalian host [5,6,7]. The parasite adopts four main forms during its life cycle: two in the insect vector, which are epimastigotes (replicative form) and metacyclic trypomastigotes (non-replicative and infectious form) [5,6,7]. The mammalian host also adopts two forms, amastigote (replicative intracellular form) and blood trypomastigote (non-replicative form) [5,6,7]. During this digenetic life cycle, the parasite faces a variety of changes in environmental conditions, and it must quickly adapt in order to survive and proliferate. The adaptation to those changes must be mediated by cell signaling pathways which can coordinate the cellular responses to different environments [8,9,10]. There are several environmental changes to which the parasite must adapt including nutrient availability, ionic composition, pH, osmolarity, temperature, oxidative stress, cell contact with host cells and tissues, host immune response and intracellular life. The cell signaling pathways in *T. cruzi* are not well known yet, although cellular signaling pathways involving second messengers such as lipids, calcium and cyclic AMP have been described [8,9,10]. 

In eukaryotes, cell signals are mostly transduced by protein kinases, which mediate the phosphorylation of key proteins directly involved in those processes. Adaptive responses are quickly regulated by reversible phosphorylation and dephosphorylation of key functional proteins, therefore, protein kinases and protein phosphatases are in the process of cell signaling. In higher eukaryotes, protein kinases can phosphorylate and activate specific transcriptional activators, which can bind to the gene promoters of target genes and activate gene expression. Trypanosomatids are unable to regulate gene expression at the transcriptional level, thus, regulation of protein function must be done at the post-transcriptional level, such as mRNA processing, translational or posttranslational levels [11,12]. Therefore, phosphorylation and dephosphorylation of key proteins must be one of the main mechanisms to regulate adaptive responses to the extra and intracellular signals. 

Oxidative stress is one of the environmental changes that must face *T. cruzi* during macrophage invasion since reactive oxygen species (ROS) and reactive nitrogen species (RNS) are produced in that process. The ROS and RNS might be able to kill the invading parasite; however, *T. cruzi* has evolved several mechanisms to escape from killing [13,14]. ROS can damage mainly proteins, lipids, and DNA from the parasite. DNA damage is critical for the parasite to replicate and survive; however, the parasite possesses a DNA repair system, which allows it to repair and replicate its DNA [13]. One of those systems to repair the kinetoplast DNA (kDNA) is through the DNA polymerase β (Tcpolβ) which is able to repair the kDNA damage caused by oxidative lesions [15,16,17]. Tcpolβ possesses intrinsic 5′-deoxyribose phosphate (dRP) lyase activity, which contributes to the kDNA repair of oxidative lesions through the base excision repair (BER) system [15,16]. Overexpression of Tcpolβ into the parasite confers resistance to high doses of hydrogen peroxide and also parasites exposed to high doses of hydrogen peroxide respond to overexpressing Tcpolβ [18]. Notably, Tcpolβ exists in two forms in cellular extracts, one that is phosphorylated (H form) and another which is not (L form). The phosphorylated form is active in DNA synthesis, while the dephosphorylated form is almost inactive [18]. 

In an earlier work, we identified predicted consensus phosphorylation sites for casein kinase 1 and 2 (CK1 and CK2), and protein kinase A and C (PKA and PKC) in Tcpolβ [17]. In recent work by using bioinformatic tools, we were able to detect consensus phosphorylation sites for CK1, CK2 and aurora kinase (AURK) in this enzyme [19]. By searching the *T. cruzi* genome database using the BLASTP tool, we found orthologues for those three protein kinases, and we cloned, expressed, and purified those enzymes. We found that recombinant *T. cruzi* CK1, CK2 and AURK were able to in vitro phosphorylate Tcpolβ and the phosphorylation by those protein kinases was able to stimulate Tcpolβ in in vitro DNA synthesis [19]. In order to gain insights into the in vivo phosphorylation of Tcpolβ, we have purified this DNA polymerase from epimastigotes cell extracts by affinity chromatography, and we identified the phosphorylation sites on the H form of Tcpolβ from epimastigotes cells. Our results show that Tcpolβ is phosphorylated in vivo by a tyrosine kinase, PKC and CK2. Furthermore, by searching the *T. cruzi* genome database using the BLASTP tool, we found three putative PKC isoforms that we have named TcPKC1, TcPKC2 and TcPKC3. We cloned the gene encoding those polypeptides, expressed and purified TcPKC1 and TcPKC2 in a recombinant form. Both isoforms were able to phosphorylate in vitro Tcpolβ, indicating that those polypeptides could be the orthologues of higher eukaryotes PKC. 

## 2. Materials and Methods

### 2.1. Immunoaffinity Purification of Tcpolβ

Native Tcpolβ was purified from epimastigotes cell extracts of the Y strain (discrete typing unit TcII) [20]. Epimastigotes were grown in liver infusion tryptose serum medium (LIT) [21] to a mid-log phase and then collected by centrifugation. The cell pellet was resuspended in a lysis buffer (100 mM Tris pH 7.9, 0.5 M NaCl, 0.1% *v*/*v* of Triton X-100 and NP-40, 0.01% *w*/*v* of SDS, 0.1 mM PMSF. A tablet of Phospho-Stop (Roche, Basel, Switzerland) was added for each 10 mL of lysis buffer. Broken cells were mildly sonicated to break the kinetoplast and nuclear DNA and the mix was centrifuged at 10,000× *g* for 30 min. The supernatant was taken and stored to −80 °C until use. Typically, each cell extract had a protein concentration of 5 mg/mL as determined by the Bio-Rad protein assay (Bio-Rad, Hercules, CA, USA). Immunoaffinity resin was prepared by incubating 2 mL of settled protein A-magnetic beads (Genscript, Piscatawey, NJ, USA) with 4 mL of anti-Tcpolβ rabbit serum for 2 h at room temperature in a disposable 10 mL column (Bio-Rad, Hercules, CA, USA). After that, the resin was washed with PBS, and the bound antibodies were crosslinked to the protein A-magnetic beads by using dimethyl pymedilate (DMP) according to the protocol of reference [22]. The antibody concentration was 2 mg/mL of settled beads. To purify native Tcpolβ, 1 mL of settled beads, containing the antibodies, were mixed with 5 mL of epimastigotes cell extract and incubated with gently rocking for 2 h at room temperature into a disposable 10 mL column. After, the beads were washed with 30 mL of lysis buffer and the bound material was eluted by using 0.1 M glycine-HCl pH 2.4 buffer. Fractions of 0.5 mL were collected and quickly neutralized with 50 μL of 2 M Tris base. A 10 μL aliquot of each fraction was analyzed in a 10% SDS-PAGE gel, followed by silver staining. Those fractions containing proteins were stored at −80 °C. Each column was used twice, and we repeated the process four times. The stored fractions were mixed and concentrated in a Centricon (Merck, Darmstadt, Germany) device until a final volume of 250 μL. The concentrated fraction was then separated in a 10% SDS-PAGE gel and the proteins were visualized by Coomassie blue R-250 staining. The band corresponding to phosphorylated Tcpolβ (H form) was cut out from the gel and sent for phosphor peptide analysis at Creative Proteomics (Shirley, NY, USA). The native phosphorylated Tcpolβ was essentially pure since all identified peptides could be matched to the Tcpolβ amino acid sequence. 

### 2.2. Protein Expression and Purification

The open reading frame of TcPKC1 (ESS64742), TcPKC2 (PWV04907) and Tcwee1 (RNF24358) were recoded according to the codon usage of *Escherichia coli* and synthesized at Genscript (Piscatawey, NJ, USA). After synthesis, the cDNAs were inserted in frame in pET15b (Merck, Darmstadt, Germany) and BL21 (DE3) competent cells were transformed with those vectors. Cells were plated on Luria-Bertani/Agar/Ampicillin media and a single colony was grown in terrific broth (TB) supplemented with 0.1 mg/mL of ampicillin. Cultures were grown to OD_600_ = 0.8–1.0 and then protein expression was induced by adding 0.5 mM IPTG to the media. After 3 h the cells were pelleted, and the collected cells were processed, and each recombinant protein was purified from inclusion bodies as described by Maldonado et al. [17]. Soluble active enzymes were further purified by Ni-NTA-Agarose affinity chromatography. Bound proteins from the column were eluted with an Imidazole gradient from 50 mM to 200 mM. Fractions from the column were analyzed in a 10% SDS-PAGE gel, followed by Coomassie blue R-250 staining. Those fractions that contained the recombinant proteins were pooled and used for the experiments. Recombinant Tcpolβ was purified as described by Maldonado et al. [17]. Mutant Tyr35Ala of Tcpolβ was made by changing the Tyr35 encoding codon to an Ala encoding codon. The recombinant mutant protein was purified according to Maldonado et al. [17].

### 2.3. Phosphorylation Assays

The assay for TcPKCs activity was carried out in 20 mM Tris pH 7.9, 10 mM MgCl_2_, 0.6 mM CaCl_2_, 4 μCi ATP, 2 mM DTT and 2% *v*/*v* glycerol in a final volume of 20 μL. Incubations were performed for 20 min at 28 °C. The amounts of each protein kinase and Tcpolβ substrate were indicated in each figure legend. Tcwee1 kinase assay was done in 50 mM Tris pH 7.9, 10 mM MgCl_2_, 10 mM MnCl_2_ and the same amount of ATP as above in a final volume of 20 μL. Incubations were performed for 30 min at 28 °C. Reactions were terminated by the addition of Laemmli loading buffer and separated in a 10% SDS-PAGE gel. After electrophoresis was completed, the gels were exposed overnight to X-ray films. The films were processed and scanned, and the signals were quantified using the Image J software (NIH, Bethesda, MD, USA).

### 2.4. Phosphoprotein Staining

In gel phosphoprotein staining was performed by using a kit from ABP Biosciences (Rockville, MD, USA) according to the manufacturer’s instructions. Fluorescence was detected by placing the gel over a UV transilluminator and the image was recorded using a digital camera.

### 2.5. Statistics

Statistical analyses were performed using the software GraphPad Prism 9 (GraphPad Software Inc., San Diego, CA, USA). Results were expressed as the mean ± standard deviation (SD). Experiments were repeated at least three times. Signal quantification on the autoradiography films was performed using the Image J software (NIH, Bethesda, MD, USA). The Shapiro-Wilk test evaluated data distribution, and differences between experimental groups were assessed using the *t*-student test. *p* < 0.05 was considered statistically significant.

## 3. Results

### 3.1. Immunoaffinity Purification of Tcpolβ from Epimastigotes Cell Extracts

By using a column containing anti-Tcpolβ antibodies crosslinked to protein A-magnetic beads, we isolated the phosphorylated H form from epimastigotes cell extracts. Analysis of the bound polypeptides by SDS-PAGE followed by silver staining showed that eluates contained several polypeptides (Figure 1A). Analysis of those polypeptides by western blot analysis revealed that one of those polypeptides was Tcpolβ H form (Figure 1C). The Tcpolβ L form was not retained in the affinity column since it was not seen in the western blot analysis. The crude extract used as an input (control load) contains both H and L forms of The Tcpolβ (Figure 1C). The bound Tcpolβ H form was stained by phosphoprotein staining, indicating that it is a “bona fide” phosphoprotein (Figure 1B). Several other polypeptides copurified with Tcpolβ (Figure 1A), and they might be Tcpolβ-associated polypeptides or just simple contaminants. It can be also observed that two proteolytic products of Tcpolβ are produced during the purification process since they react in the western blot analysis. We have not studied further those polypeptides which copurified in the affinity column. 

### 3.2. Phosphorylation Sites on Tcpolβ H Form

In order to identify the phosphorylation sites and the protein kinases that are able to phosphorylate the amino acid residues on Tcpolβ, the H form was excised from the gel, treated with two different proteases and phosphor peptides were analyzed by mass spectrometry (Creative Proteomics, see Section 2). The results indicate that all peptides matched the Tcpolβ amino acid sequence, and this DNA polymerase was heavily phosphorylated at Tyr35, Thr123, Thr137 and Ser286 (Figure 2). Unfortunately, we were unable to get peptide coverage at the CK2 regulatory domain of Tcpolβ, where there exist several phosphorylation consensus sites for CK2 and presumably Tcpolβ is in vivo phosphorylated at some of those sites as well.

### 3.3. Protein Kinases Can Phosphorylate Tcpolβ In Vivo

By using the NetPhos 3.1 server that predicts Ser, Thr and Tyr phosphorylation sites in eukaryotic proteins, we found that Tyr35 is predicted to be phosphorylated by an insulin receptor kinase (INSR); however, we were unable to find orthologous genes encoding the INSR in the *T. cruzi* genome. Moreover, although Tyr phosphorylation exists in *T. cruzi*, typical Tyr protein kinase encoding genes have not been found yet in this parasite. This indicates that perhaps Tyr35 is phosphorylated by a dual protein kinase that can phosphorylate Ser/Thr and Tyr residues as well. Thr 123 is a phosphorylation site for CK2, and in a previous report, we showed that Tcpolβ is in vitro phosphorylated by CK2 [19]. Thr137 and Ser286 are phosphorylation sites for PKC. PKC-like activities have been previously reported in both *T. cruzi* and *T. brucei*; however, the genes encoding those polypeptides have not been identified yet [22,23,24,25].

### 3.4. PKC-like Enzymes Encoded in the T. cruzi Genome

Next, we sought to identify orthologues of PKC in the *T. cruzi* genome database, by searching using the BLASTP tool and querying with the amino acid sequence of human PKCα and PKCβ isoforms. Several polypeptides with high identities in the catalytic kinase domain were found and those belong to the protein kinase A, G and C family (AGC family); however, a high number belong to the catalytic subunit of the PKA-like subfamily, which is regulated by cAMP levels in the cell. Notably, we identified three polypeptides, whose amino acid sequences have a high identity to the catalytic kinase domain of the AGC family (Appendix A) but contain N-terminal domains which could bind phospho inositol phosphates (PIPs) (Figure 3). We have named those polypeptides as TcPKC1, TcPKC2, and TcPKC3, and most likely they are orthologues of higher eukaryotes PKC, although the regulatory domains have considerably diverged (Figure 3). Those polypeptides have homologs in *T. brucei* as well (Appendix A). We will describe and further discuss those domains in the Discussion section. 

### 3.5. PKC-like Enzymes from T. cruzi Can Phosphorylate Tcpolβ In Vitro

As described above, to identify the in vivo phosphorylation sites of Tcpolβ we purified the enzyme from epimastigotes cell extracts and determined the phosphorylation sites. We found that PKC and CK2 can in vivo phosphorylate Tcpolβ, in addition to an unknown tyrosine kinase. As mentioned, PKC encoding genes have not been cloned and characterized in *T. cruzi* so far, and therefore, we identified orthologues of mammalian PKCs in the *T. cruzi* genome. We found three major PKCs in *T. cruzi*, named TcPKC1, TcPKC2 and TcPKC3, although we do not rule out the possibility that other isoforms could exist in the parasite. Both TcPKC1 and TcPKC2 can in vitro phosphorylate Tcpolβ in the absence of DAG and calcium. We have not cloned and characterized further TcPKC3; however, we believe that most likely is also able to phosphorylate Tcpolβ.

In order to determine whether those identified polypeptides could phosphorylate in vitro Tcpolβ, which has two phosphorylation sites for PKC, we cloned, expressed and purified TcPKC1 and TcPKC2. Analysis of the purified polypeptides by SDS-PAGE, followed by Coomassie Blue staining shows that the obtained protein kinases were essentially pure (Figure 4A). TcPKC1 has an apparent MW of 120 KDa, whereas TcPKC2 has an apparent MW of 90 KDa. The recombinant pure PKCs were used to in vitro phosphorylate Tcpolβ. The results show that both PKC isoforms are able to in vitro phosphorylate Tcpolβ in a dose-dependent fashion (Figure 4B,C). Moreover, TcPKC1 is able to get weakly auto-phosphorylated in the kinase reaction (Figure 4B); however, TcPKC2 was not auto-phosphorylated under those conditions (Figure 4B). When increasing amounts of TcPKC1 were assayed in the absence of Tcpolβ a higher autophosphorylation signal was obtained (Figure 4D,E). Neither TcPKC1 nor TcPKC2 was able to phosphorylate protamine sulfate, which is a typical substrate of higher eukaryotes PKCs, indicating that TcPKCs might have a slightly different substrate specificity (data not shown). On the other hand, it is known that higher eukaryotes PKCs require calcium for full activity and possess calcium-binding domains. We did not detect any calcium-binding domains in the *T. cruzi* PKCs by using bioinformatic analysis and according to this neither TcPKC1 nor TcPKC2 required calcium for kinase activity, since the phosphorylation of Tcpolβ can occur in the absence of calcium and in the presence of EGTA (Figure 4F). Furthermore, both TcPKC1 and TcPKC2 were not inhibited by Gouml 6983 (data not shown), which is a selective and potent inhibitor of mammalian PKCs, which again indicates that TcPKCs have some slightly different substrate specificity compared with the mammalian counterparts.

### 3.6. Tyrosine Phosphorylation of Tcpolβ

As mentioned earlier, we were unable to find any specific tyrosine kinase encoding gene in the *T. cruzi* genome, although we found that Tcpolβ is in vivo phosphorylated by a tyrosine kinase. We believe that Tcpolβ is phosphorylated by a dual protein kinase. In this regard, Wee1 is a protein kinase from *Schizosaccaromyces pombe* that has dual activity with Ser/Thr and Tyr phosphorylating activities [26]. By querying with *S. pombe* Wee1 amino acid sequence we were able to find several wee1-like polypeptides encoded in the *T. cruzi* genome (Appendix A). We chose one of them to be further studied to investigate whether a Wee1-like kinase could phosphorylate in vitro Tcpolβ. We named the gene encoding the *T. cruzi* Wee1-like as Tcwee1 and it was cloned, expressed and the polypeptide was purified. The Tcwee1 protein has an apparent MW of 50 kDa and it is nearly pure after purification as judged by SDS-PAGE, followed by silver staining (Figure 5A). The results showed that Tcwee1 was able to phosphorylate Tcpolβ in a dose dependent manner, indicating that Tcpolβ is a substrate for Tcwee1 for in vitro phosphorylation (Figure 5B). We were interested to investigate whether a mutation of Tyr35 in Tcpolβ, which is in vivo phosphorylated, can abrogate Tcwee1 phosphorylation. Tyr35 lies in the lyase domain of Tcpolβ [27] and the substitution by Ala does not affect the DNA synthesis activity of this DNA polymerase (data not shown). Recombinant Tcpolβ is active in DNA synthesis, although requires high amounts to synthetize DNA [17]. The mutation Tyr35Ala does not abrogate in vitro phosphorylation of Tcpolβ since the levels of phosphorylation do not vary by the mutation (data not shown). 

However, we still cannot rule out the possibility that Tcwee1 could phosphorylate Tcpolβ at Tyr35, since additional Tyr as well as Ser/Thr residues exist in Tcpolβ that could be in vitro phosphorylated by Tcwee1. Nevertheless, we have identified another protein kinase that belongs to the Wee1 family in *T. cruzi*, which is able to in vitro phosphorylate Tcpolβ and it might be also able to phosphorylate in vivo Tcpolβ, playing important roles in the regulation of the *T. cruzi* cell cycle. 

## 4. Discussion

In this work, we have presented evidence that Tcpolβ H form is in vivo phosphorylated by CK2, PKC-like kinases and an unknown Tyr kinase. We also described three putative PKC isoforms from *T. cruzi*, which can in vitro phosphorylate Tcpolβ. Also, we have found that Tcpolβ can be in vitro phosphorylated by protein kinases of the Wee1 family. Those results are very important to understand the signal transduction processes in *T. cruzi* since most of the metabolic routes and their components are still poorly understood.

Signal transduction pathways play a key role in regulating important biologic processes in both unicellular and multicellular organisms. The signal transduction pathways largely can control the manner in which cells respond to a given stimulus and are responsible to coordinate the functions of different cell types in animals [28,29,30]. They are also responsible for controlling cell growth and differentiation in unicellular organisms [31,32]. Intracellular signal transduction pathways are mostly activated by second messenger molecules, which include cyclic AMP, cyclic GMP, calcium, nitric oxide and lipophilic second messengers such as diacylglycerol (DAG), ceramides and PIPs [33]. Unicellular organisms can also respond to environmental stimuli through the activation of signal transduction pathways [31,32]. As mentioned earlier, in many organisms reversible protein phosphorylation is one of the most important mechanisms for the regulation of adaptive responses to intra and extracellular signals. Phosphorylation of key target proteins alters the activities of those and is carried out by activated protein kinases and dephosphorylation of those proteins is performed by protein phosphatases [34,35]. As mentioned earlier, *T. cruzi* has a complex life cycle and presents four morphogenic stages, in which each development stage is distinguished morphologically and shows stage-specific differences in surface antigens and intracellular components. Major signal transduction pathways have been described in *T. cruzi* and those are reviewed in references [9,10,32]. The cyclic AMP-dependent pathway, involving PKA, and the mitogen-activated protein kinase (MAPK) pathway have been described with some details [9,10,32]. The cyclic AMP is a common second messenger that plays important roles in cell growth and differentiation. One of the typical effectors of cyclic AMP is PKA, which is present in *T. cruzi*, and also cyclic AMP can mediate *T. cruzi* differentiation [9,10,36,37]. On the other hand, the MAPK pathway plays key roles in regulating cell proliferation, differentiation, stress response and apoptosis. A MAPK homolog, named TcMAPK2, has been cloned and characterized in *T. cruzi* [9]. This polypeptide is catalytically active, and it is located in the cytoplasm of epimastigotes, while in trypomastigotes is located along the flagella and on the plasma membrane of intracellular amastigotes [9]. Huang has identified thirteen different TcMAPKs encoding genes; however, the genes and encoded proteins have not been characterized yet [9].

It has been demonstrated that phorbol esters and cell-permeable DAG can stimulate the metacyclogenesis of *T. cruzi* [38]. Also, free fatty acids present in the intestinal extracts of *Triatoma infestams* can induce cell differentiation of *T. cruzi* epimastigotes to the metacyclic infective metacyclic form [25]. Also, oleic acid can trigger a transient calcium signal in epimastigotes, which is necessary for *T. cruzi* differentiation into the metacyclic form [25]. It has been proposed that parasite differentiation is the result of PKC activation since phorbol esters, DAG together with phosphatidylserine and calcium are potent activators of PKC. Also, free fatty acids can directly stimulate in vitro *T. cruzi* PKC activity [25]. Earlier work by Gomez et al. has partially characterized PKC activity from *T. cruzi* epimastigotes cell extracts [22,23]. The PKC activity from epimastigotes preferentially phosphorylated histone H1, and it was stimulated by calcium, phorbol esters and DAG [22,23]. The kinase activity was associated with microsomal and cytosolic fractions. Those PKC activities were able to react with antibodies against the human PKCα isoform which has an apparent MW of 80 kDa in epimastigotes-derived fractions and in trypomastigote cell extracts [22,23]. Immunoaffinity purification of the PKC from epimastigotes-derived fractions resulted in the isolation of an 80 kDa polypeptide, which undergoes autophosphorylation [22]. However, this polypeptide has not been characterized further. Notably, TcPKC1 undergoes autophosphorylation; however, the MW of TcPKC1 and TCPKC2 are higher than the 80 kDa polypeptide described and they do not require Ca for activity. We do not have an explanation for this difference, but it might be possible that the 80 kDa polypeptide might be associated with membrane proteins, which could impose an additional requirement for Ca and phosphatidylserine and DAG. On the other hand, TcPKC1 might undergo in vivo processing by proteolysis of the lipocalin domain. Belaunzarán et al. by using antibodies determined the presence of several PKC isoforms, including classical and novel forms which were differentially expressed in the different *T. cruzi* stages [25].

Amino acid sequence analysis of those protein kinases revealed that all three contain a conserved kinase catalytic domain at the C-terminus of the polypeptide, while the N-terminus contains regulatory domains, which can bind different regulatory ligands. TcPKC1 has 858 amino acid residues and a Phox (PX) homology domain located from amino acid 37 to 147 that is predicted to bind phosphoinositide (acidic phospholipids) [39], while at the C-terminal end contains a lipocalin-like domain (711–855), which could bind free fatty acids [40]. TcPKC2 is a 705 amino acid residues kinase, which has a Zinc-finger domain of the FYVE type at the N-terminus, which spans from residue 86 to 143, and the typical kinase catalytic domain at the C-terminus. This type of Zinc-finger domain is predicted to bind phospholipids [41]. No other recognizable domains are present in TcPKC2. On the other hand, TcPKC3 is smaller than the others (458 amino acid residues) and possesses a pleckstrin homology (PH) domain at the N-terminal region spanning from residue 3–98 and at the C-terminus has the kinase catalytic domain. The PH domain is predicted to bind phospholipids or mediate protein-protein interactions with proteins involved in signal transduction [42]. The C-terminal end has a STKX domain (397–438), present in some Ser/Thr kinases. We did not find the classical conserved regulatory domains found in higher eukaryotes PKCs, such as calcium-binding domain, DAG and phorbol esters binding domains. The recombinant *T. cruzi* PKCs are not stimulated by calcium since they can work in the absence of calcium and in the presence of the calcium chelator EGTA. Whether those recombinant polypeptides are stimulated by DAG and phosphatidylserine, phorbol esters or free fatty acids is still unknown, but we believe that stimulation by those compounds might be possible. It is noteworthy that the mammalian PKC conserved region 2 (C2) also binds phosphoinositide [41,43,44]. In conclusion, we have described a PKC-like signal transduction pathway in *T. cruzi,* and we have identified and characterized three PKC isoforms from *T. cruzi* that might be able to mediate signal transduction pathways through second messengers, which could be involved in metacyclogenesis, a differentiation process of *T. cruzi* epimastigote forms. Besides, it seems to be that those TcPKCs can regulate the DNA synthesis activity of Tcpolβ, which is involved in DNA replication and kDNA repair.

Very little is known regarding Tyr phosphorylation in *T. cruzi*. It is known that Tyr phosphorylation exists in *T. cruzi* and *T. brucei*; however, the protein kinases involved in this process have not been identified and we were unable to find typical Tyr kinases encoded in the *T. cruzi* genome [45,46,47,48,49]. Studies carried out by Marchini et al. [47] on the analysis of the *T. cruzi* phosphoproteome have revealed that the distribution of phosphorylated residues was 84.1% on Ser, 14.9% on Thr and 1.0% on Tyr. Tyrosine phosphorylation is known to play a key role in cell signaling pathways, and it was found that several protein kinases were found to be phosphorylated on tyrosine residues, especially on activation loops [47]. Notably, Tcpolβ is in vivo phosphorylated at Tyr35 and the NetPhos 3.1 program predicts that is a phosphorylation site for insulin receptor (INSR) or an epidermal growth receptor kinase (EGF), but orthologues of those kinase receptors are not present in the *T. cruzi* genome. We believe that Tyr35 in Tcpolβ can be phosphorylated by a dual Ser/Thr and Tyr protein kinase [50]. *S. pombe* Wee1 is one of those dual kinases [26]. For such a reason, we identified a Wee1-like kinase in *T. cruzi*. There are several protein kinases with homology to Wee1 kinase in *T. cruzi* and we choose one of them, based on the percent of identity with *S. pombe* Wee1. We cloned and characterized the encoding gene and named the polypeptide as Tcwee1, which is 420 amino acid residues long protein. The pure recombinant Tcwee1 is able to in vitro phosphorylate Tcpolβ, but also can phosphorylate a mutant Tyr35 Tcpolβ. This indicates that Tyr35 is not a target for Tcwee1 or alternatively this kinase can phosphorylate other Ser/Thr and Tyr residues in addition to Tyr35 since Tcpolβ is rich in Ser, Thr and Tyr that can be phosphorylated by Tcwee1.

Recently, studies of quantitative proteomics and phosphoproteomics of *T. cruzi* epimastigotes cell cycle have revealed that several protein kinases are involved during cell cycle progression. Those protein kinases predicted to act in regulated phosphopeptide sites are: PKA, cdc2, DNAPK, CK2, PKC and p38MAPK [51]. Particularly, PKC shows to be upregulated at 6 h, indicating that it might play a role during cell cycle progression. It is possible that one of the regulated proteins by PKC is Tcpolβ since it plays a key role in kDNA repair and replication [51]. In *T. cruzi,* PKC-like enzymes might become activated by Ca and lipids released by different cell signals and in turn could phosphorylate Tcpolβ to modulate its function and regulate kDNA replication or metacyclogenesis. On the other hand, CK2 is known to regulate the control of cell proliferation and it is also involved in signal transduction and influences several biological processes. It is possible that might be able to translocate to the kinetoplast and Tcpolβ could be one of the key targets of CK2 to regulate the activity of Tcpolβ and in such a manner could control kDNA repair and replication. Regarding Tcwee1, this dual kinase has been shown to have a role in the proper timing of cell division controlling the entry in mitosis and the DNA replication in the S phase. We can speculate that Tcwee1 can regulate Tcpolβ activity and in such a way Tcwee1 could control the kDNA replication. Additionally, Tcwee1 may phosphorylate nuclear components required for DNA replication and modulate the cell cycle of *T. cruzi*.

Lastly, protein kinases represent promising drug targets for several human and animal diseases. The kinome for *T. cruzi* has been defined and those protein kinases most likely are involved in regulating cell cycle control, differentiation, and response to stress and signals during the complex life cycle of the parasite. Therefore, the differences between the parasite and mammalian protein kinases should be exploited to develop novel anti-parasitic chemotherapeutic agents. It seems to be that the PKC signaling pathway in *T. cruzi* is significantly different as compared with mammals. Thus, it might be possible to design specific inhibitors for the parasite PKC, therefore the parasite will be killed without harming the host. This should be exploited to develop new drugs to treat Chagas disease.

## Figures and Tables

**Figure 1 cells-11-03693-f001:**
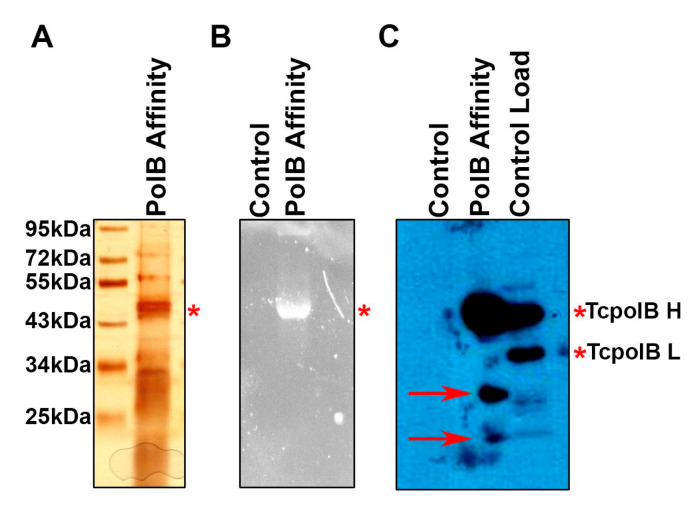
Purification and analysis of Tcpolβ. Cell extracts from epimastigotes cells were subjected to antibody affinity chromatography and bound proteins were eluted with Glycine-HCl (pH 2.5) buffer. Eluates were concentrated using Centricon filters and subjected to SDS-PAGE (10% gel), and analyzed by silver stain (**A**), phosphoprotein staining (**B**) and western blot with anti-Tcpolβ antibodies (**C**). The H and L forms of Tcpolβ are indicated. Proteolytic peptides of Tcpolβ are indicated by red arrows in panel (**C**). * indicate the position of TcpolB H.

**Figure 2 cells-11-03693-f002:**
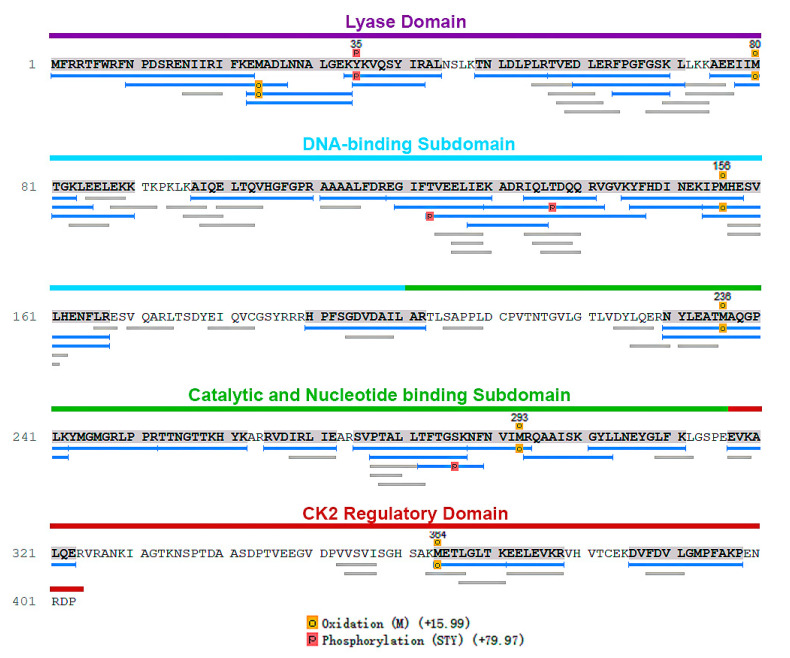
Identification of phosphor sites on Tcpolβ. Tcpolβ H form was purified from epimastigotes cell extracts by antibody affinity chromatography. The concentrated eluates were subjected to SDS-PAGE (10% gel) and Coomassie Blue R-250 staining. The band corresponding to Tcpolβ H form (20 μg) was cut out from the gel and treated with two proteases. The phosphor peptides were analyzed by LC-MS/MS spectrometry at Creative proteomics. The sequenced peptides are shown and the position of phosphorylated Ser, Tyr and Thr residues are shown. The different domains of Tcpolβ are shown over the amino acid sequence. It can be observed that in the middle and at the CK2 regulatory domain of the polypeptide there was not good peptide coverage, therefore some in vivo phosphorylation sites might be missing from the analysis. The Tyr35 can be phosphorylated by an unidentified dual protein kinase. Thr123 is phosphorylated by CK2. Thr137 and Ser286 are phosphorylated by PKC.

**Figure 3 cells-11-03693-f003:**
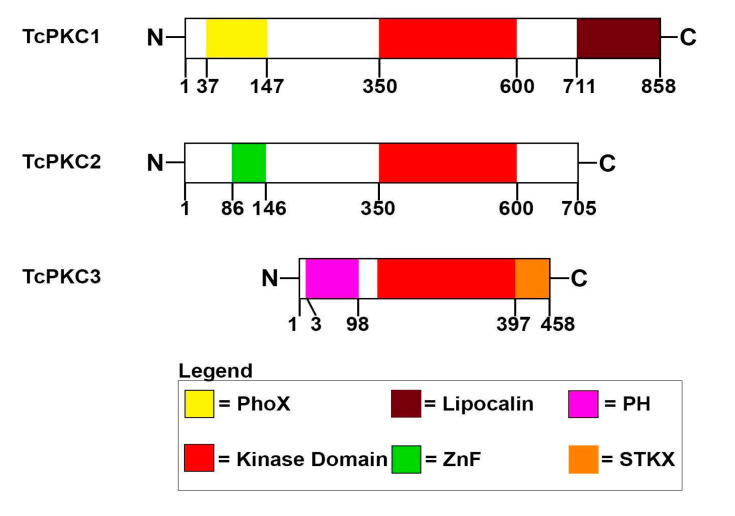
Schematic representation of TcPKC1, TcPKC2 and TcPKC3. By searching the *T. cruzi* genome database, three putative PKCs were found: TcPKC1 (ESS64742), TcPKC2 (PWV04907) and TcPKC3 (XP_809943.1). Bioinformatic analysis indicated that those polypeptides contained different domains, which are indicated at the bottom of the figure. PhoX, PH and the ZnF domain are predicted to bind PIPs. The kinase domain is the catalytic domain found in the AGC protein kinases family. The lipocalin domain is predicted to bind free fatty acids, whereas the STKX domain is found in some protein kinases.

**Figure 4 cells-11-03693-f004:**
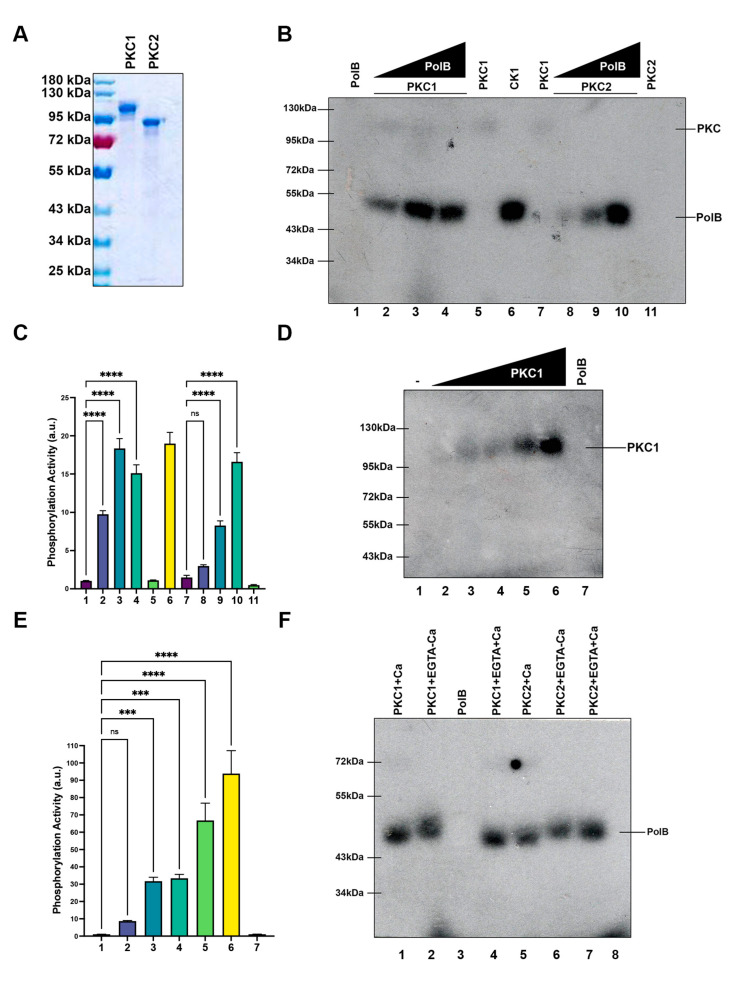
TcPKC1 and TcPKC2 can phosphorylate Tcpolβ in vitro. The genes for both TcPKC1 and TcPKC2 were cloned, expressed and the proteins purified by NTA-Ni-Agarose column. (**A**) Two hundred (200 ng) of each purified protein were analyzed in a PAGE-SDS (7% gel) followed by Coomassie Blue R-250 staining. (**B**) Tcpolβ phosphorylation by TcPKC1 and TcPKC2. Two (2 pmol) of each protein kinase were used to phosphorylate 1, 2 and 4 pmol of recombinant Tcpolβ. In the figure can be seen that TcPKC1 is weakly auto phosphorylated (lanes labeled PKC1). The phosphorylated Tcpolβ (PolB) is shown in the figure. A positive control is shown using CK1 (lane labeled CK1). Tcpolβ by itself is not phosphorylated (lane labeled PolB). (**C**) Quantification of (**B**). (**D**) TcPKC1 is auto phosphorylated. Several amounts (2, 4, 8, 16 and 32 pmol) of TcPKC1 (PKC1) were incubated in the kinase reaction buffer. Tcpolβ alone (PolB) was used as a negative control. (**E**) Quantification of (**D**). (**F**) TcPKC1 and TcPKC2 do not require calcium for activity. Four (4 pmol) pmol of each protein kinase was mixed with 4 pmol of Tcpolβ in the presence of EGTA (2 mM) without Ca^+2^ or in the presence of Ca (0.6 mM) and excess of EGTA (2 mM) as indicated at the top of the figure. Tcpolβ (PolB) was included as negative control. The slight difference in the mobility of phosphorylated Tcpolβ is due to the presence of Ca^+2^ and EGTA. n.s. indicate non-significance differences between groups; *** = indicates *p* < 0.001; **** = indicates *p* < 0.0001.

**Figure 5 cells-11-03693-f005:**
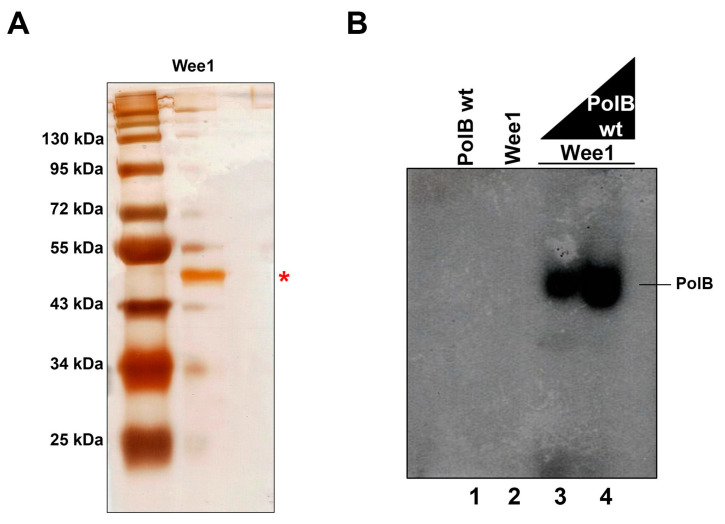
Tcwee1 phosphorylates Tcpolβ. The gene encoding a Tcwee1-like kinase was cloned, expressed and the protein purified. (**A**). Fifty ng (50 ng) of purified Tcwee1 was analyzed in a 10% SDS-PAGE followed by silver staining. (**B**) Two pmol (2 pmol) of Tcwee1 was added to reactions containing 2 and 4 pmol of Tcpolβ. Tcwee1 by itself does not get auto phosphorylated under the conditions used in the reaction (see Materials and Methods); however, we do not rule out the possibility that Tcwee1 under different conditions might get auto phosphorylated, Tcpolβ alone was used as a negative control. * indicates the position of Tcpolβ protein.

## Data Availability

Not applicable.

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
