# Peer review of "Trypanosoma cruzi DNA Polymerase β Is Phosphorylated In Vivo and In Vitro by Protein Kinase C (PKC) and Casein Kinase 2 (CK2)"

_cells, 2022, doi:10.3390/cells11223693_

Round 1
Reviewer 1 Report
1. In the methodology it is necessary to state which statistical analysis was performed. Which method? In which program? What p value was used?
2. The results on page 4 are written and bold. Please remove bold.
3. If possible, increase the legend of figure 2.
4. The discussion text on page 11 and lines 392 to line 402 is very well written and perfectly details the results obtained. I suggest inserting this text into the results.
5. This same text should be simplified in the discussion just to support the comparison with the literature.
6. It is not used to place in the text the figure that corresponds to the discussion. Remove from line 404 (see figure 3)
7. Review the formatting of bibliographic references. Some came out with the year in bold and others did not.
Author Response
We would like to thank both reviewers for their comments on the manuscript. Those comments will certainly improve its quality. We hope that now it would be ready for publication in Cells. If you have any further queries, please do not hesitate to contact us.
Best Regards,
Edio Maldonado
- In the methodology it is necessary to state which statistical analysis was performed. Which method? In which program? What p-value was used?
RESPONSE: A new paragraph was added to the methodology section with the required information.
- The results on page 4 are written and bold. Please remove bold.
RESPONSE: The bold was removed in page 4. Thank you for the comment.
- If possible, increase the legend of figure 2.
RESPONSE: The legend of figure 2 was increased.
- The discussion text on page 11 and lines 392 to line 402 is very well written and perfectly details the results obtained. I suggest inserting this text into the results.
RESPONSE: Thank you for the comments. The text was moved to the results section.
- This same text should be simplified in the discussion just to support the comparison with the literature.
RESPONSE: The discussion section was improved according to the referee’s comments.
- It is not used to place in the text the figure that corresponds to the discussion. Remove from line 404 (see figure 3)
RESPONSE: Line 404 was removed.
- Review the formatting of bibliographic references. Some came out with the year in bold and others did not.
RESPONSE: Bibliography was corrected.
Reviewer 2 Report
In this work, the authors report the purification of the H form of this DNA polymerase and identified its phosphorylation sites. The authors argue that DNA polymerase β is in vivo phosphorylated at several amino acid residues including Tyr35, Thr123, Thr137 and Ser286 and that Thr123 is phosphorylated by casein kinase 2 and Thr137 and Ser286 are phosphorylated by protein kinase C-like enzymes. The protein kinase C encoding genes were identified and genes were cloned, expressed in bacteria and purified. The results suggest that T. cruzi possesses three different protein kinase C-like enzymes (TcPKC1, TcPKC2, and TcPKC3) being the two initial ones able to in vitro phosphorylate recombinant DNA polymerase β. They additionally found that TcPKC1 gets auto phosphorylated. The authors also identified a Wee1 homolog in T. cruzi and demonstrated that the recombinant kinase was able to in vitro phosphorylate recombinant DNA polymerase β, but have doubts of if there is a specific phosphorylation on Tyr35.
The manuscript herein presented is really interesting and presenting data fundamental to understand the cell signalling pathways involving PKC-like kinases in T. cruzi as well as the holistic comprehension of the hole kinome of the parasite.
The methods are thoroughly described, are methodologically correct and are adequate to address the aims established by the authors The statistical analysis is correct. Tables and figures are adequate and present the data in a very thorough way.
The manuscript is very well described and organized and the data is very good and thoroughly presented. Still, a small query arises:
What can be the functional meaning for Trypanosoma cruzi DNA polymerase β of being phosphorylated by an ubiquitous enzyme (like is the case of Casein kinase 2), as well as by more specific ones like the lipid sensitive Protein kinase C or even more, the Wee1 like protein kinase that play critical roles in the proper timing of cell division and controlling entry into mitosis and DNA replication during S phase.
This is even more important since the final the conclusion says that: It seems to be that the PKC signalling pathway in T. cruzi is significantly different as compared with mammals. Thus, it might be possible to design specific inhibitors for the parasite PKC, therefore the parasite will be killed without harming the host. This should be exploited to develop new drugs to treat Chagas disease.
For all these reasons I recommend to accept this manuscript after the thoughts of the authors in this regard are included.
Author Response
We would like to thank both reviewers for their comments on the manuscript. Those comments will certainly improve its quality. We hope that now it would be ready for publication in Cells. If you have any further queries, please do not hesitate to contact us.
Best Regards,
Edio Maldonado
Referee 2
What can be the functional meaning for Trypanosoma cruzi DNA polymerase β of being phosphorylated by an ubiquitous enzyme (like is the case of Casein kinase 2), as well as by more specific ones like the lipid sensitive Protein kinase C or even more, the Wee1 like protein kinase that play critical roles in the proper timing of cell division and controlling entry into mitosis and DNA replication during S phase.
RESPONSE: A new paragraph was added to the Discussion section, regarding PKC, CK2 and Wee1.